# Coping with Symptoms of Mental Health Disorders among University Students during the COVID-19 Pandemic in Relation to Their Lifestyle Habits

**DOI:** 10.3390/medicina59010180

**Published:** 2023-01-16

**Authors:** Tamara Jovanović, Aleksandar Višnjić

**Affiliations:** 1Faculty of Medicine, University of Niš, 18000 Niš, Serbia; 2Institute of Public Health of Niš, 18000 Niš, Serbia

**Keywords:** lifestyle habits, COVID-19, depression, anxiety, stress, mental health

## Abstract

*Background and Objectives*: The time of the pandemic brought great difficulties, both in state and interstate systems, industry, trade, and with individuals themselves. In addition, numerous studies have shown a drastic increase in mental disorders in people around the world. Therefore, the basic idea of our study was to investigate these disorders in university students in relation to their different lifestyles. *Materials and Methods*: The cross-sectional study was carried out at the University of Niš (Serbia) from December 2021 to February 2022. All of the participants were assessed by using appropriate questionnaires. The study included 1400 randomly selected students (692 females and 708 males). The statistical analysis of the data included the application of multiple regression analyses and correlation tests. *Results*: Statistical analysis indicates that extremely severe levels of depression symptoms were reported by 232 students (16.6%). Severe and extremely severe anxiety symptoms were reported by 480 students (34.3%). Multiple linear regression analysis found that for the increased depressive symptoms, the “most deserving” parameters were related to the consumption of alcoholic beverages and psychoactive substances (β = 0.10, and 0.11, respectively), compared to the period before the COVID-19 pandemic. For anxiety symptoms, the main role was played by alcohol consumption (β = 0.11) but also by the use of social networks as an adequate substitute for deprived content during the pandemic (β = 0.13). Alcohol consumption was the most “responsible” for elevated stress levels compared to the period before the pandemic (β = 0.19). *Conclusions*: Due to the COVID-19 pandemic, symptoms of depression, anxiety, and stress were drastically increased in the university students. There was significantly more frequent consumption of alcoholic beverages and psychoactive substances among them. That is why social support from a close environment is the most important strategy in coping with mental health issues during emergency situations.

## 1. Introduction

The coronavirus disease, which is believed to have originated in the Chinese city of Wuhan in late 2019, was already declared a public health emergency of international concern by the World Health Organization (WHO) in January 2020, and it unexpectedly turned into a global pandemic by March 2020 [1,2]. The public health response to prevent the spread of COVID-19 was launched at various levels in all affected countries, including Serbia [3]. To reduce the risk of exposure to COVID-19, social distancing has been proposed and implemented everywhere. People were not allowed to leave their homes, except in very few situations, but even then they had to maintain physical distance. This intervention not only affected all ongoing activities but also led to a huge negative effect on people’s mental health. Fear of infection, high mortality associated with the virus, lack of appropriate therapy and the uncertain end of the epidemic, as well as constant exposure to unfavorable news, interrupted daily routine, economic losses and inability to participate in social events, contributed in many ways to a poor mental state [4]. Quarantine was an unpleasant experience for many. Separation from loved ones, loss of freedom, insecurity and boredom may be triggers for increased levels of stress, anxiety, depression and, in the most extreme cases, even suicide. The psychological stress from situations of this magnitude can have a long-lasting impact on one’s overall psychosocial well-being [5,6]. The literature on the effects of earlier population crises also shows a significant increase in the prevalence of mental disorders. For example, the prevalence of major depressive episodes doubled after the financial crisis in Hong Kong in 2008 and the financial crisis in Greece in 2009, and an increased prevalence of depressive and anxiety disorders was also observed in populations in conflict-affected countries between 2000–2017 [7,8,9].

The COVID-19 pandemic has also affected negative changes in health behaviors, such as physical activity, smoking tobacco, alcohol consumption, social media use, and sleep [10,11,12,13]. It was found that students experience a stronger psychological impact of the pandemic, as well as higher levels of stress, anxiety and depression compared to other population groups, as well as compared to the period before the epidemic [14,15,16,17]. Depression, anxiety and behavioral disorders are also among the leading causes of illness and disability among young people [18].

Research that dealt with young people, as a particularly sensitive group, from the beginning of the COVID-19 pandemic until today, have produced a lot of information about significantly lower health-related quality of life (HRQoL), damaged relationships with friends and reduced social contacts, increased number of arguments within families, difficulties with online learning and keeping up with classes, which are all triggers for mental health problems. Many of them studied the occurrence of emotional symptoms, behavioral problems, hyperactivity and problems with peers, as well as self-reported anxiety, depressive symptoms and psychosomatic complaints, such as headaches, stomach aches, feeling low or sleeping problems [19,20,21].

From a complete lockdown to recommendations related to gatherings in closed spaces, the decisions of the Government of Serbia have changed depending on the current epidemiological situation. Among the measures that were in force in Serbia at the time of conducting our research are the following: wearing protective masks indoors, maintaining distance and COVID passes for the hospitality industry after 8 p.m. In addition, what significantly influenced the student population itself was the fact that classes at most faculties were held exclusively online.

Therefore, our basic idea was to investigate the mental health symptoms of students in light of current circumstances and in relation to their lifestyle habits.

## 2. Methods

### 2.1. Participants

This cross-sectional study was carried out at the University of Niš (Serbia) from December 2021 to February 2022. All of the participants were assessed by using appropriate questionnaires. The study included 1400 randomly selected students (692 females and 708 males) of the faculties of medicine, dentistry, pharmacy, nursing, law, economy, civic and mechanic engineering, electronics, and different humanities sciences. During this period, there were a total of 24,047 students at the University of Niš, and according to the valid statistical formula, for a confidence level of 95%, the minimum sample size calculated was 379 students. This confirms that our randomized sample was very representative.

### 2.2. Procedure

The students reported their socioeconomic characteristics and lifestyle habits using self-administered questionnaires. Symptoms of depression, anxiety, and stress were assessed using the Depression Anxiety Stress Scale (DASS 42), a device for measuring psychological health [22,23]. The survey was performed in amphitheaters of all the mentioned faculties by trained assistants and was intended to last a maximum of 30 min, including the time needed for instructions.

### 2.3. Measures

The DASS42 is a set of three self-report scales for depressive symptoms, anxiety, and stress, designed to measure negative emotional states. Each of those three scales contains 14 questions, divided into subscales of 2–5 items of similar content [22,23]. The surveyed students from the University of Niš were asked to rate each item so that the number of points represents the extent to which they experienced each of the listed conditions during the week preceding the survey, from 0 (never) to 3 (mostly or almost always). Internal consistency of the entire DASS 42 scale expressed by Cronbach’s alpha coefficient was α = 0.88.

Students also answered questions related to some lifestyle habits:

How many hours do you spend on social networks on average per day? How much sleep do you get on average at night (number of hours)? How often do you consume alcohol (Likert scale from 1–6, where 1 = never, 2 = very rarely, 3 = rarely, 4 = occasionally, 5 = frequently, and 6 = very frequently)? Is it less, the same or more compared to the period before the COVID-19 pandemic? How often do you consume psychoactive substances (on Likert scale from 1–6)? Is it less, the same or more compared to the period before the COVID-19 pandemic? How many close friends would you say you have in real life? Are you in an emotional relationship (Yes or No)?-How long have you been in a relationship, or how long have you been single (number of years)? Were you under home isolation during the COVID-19 pandemic due to infection or contact with an infected person (Yes or No)? To what extent did social networks provide you with an adequate replacement for the content that was denied you during the pandemic (impossibility of gathering in catering facilities, cancellation of sports, cultural and other events, etc.) (on a scale of 1 to 10)?

### 2.4. Statistical Analysis

Statistical analysis was performed using SPSS 17.0 software (SPSS Inc., Chicago, IL, USA) in Windows 7 Ultimate. The research results are presented in tables.

The statistical analysis of the data included the application of descriptive tests, multiple linear regression analysis, and correlation tests. The descriptive statistics were performed to report the analysis of the data considering lifestyle habits of the university students that were presented as mean and standard deviations. The categorical variables were shown as frequency and percentages. Pearson and Spearman correlations were used to determine the strength of the relationships between the examined variables. Multiple linear regression analysis was used to estimate the coefficients of linear regression (β) and 95% confidence intervals (CIs) of the interactive relationships of several independent variables with depression, anxiety, and stress. The statistical significance was set at *p* < 0.05.

### 2.5. Ethics

The study procedures were carried out in accordance with the Declaration of Helsinki. All subjects were informed about the study, and all provided informed consent. The Ethics Committee of the Faculty of Medicine of the University of Niš approved the study (No. 12-6647-2/6).

## 3. Results

The study included 1400 participants, 692 female (49.4%) and 708 male students (50.6%). The mean value of the year of study was 2.73 (SD 1.58), while the mean age was 24.51 (SD 6.22)—female 24.81 (SD 8.40) and male 24.22 (SD 2.69). Other examined parameters are shown in Table 1.

Extremely severe levels of depression symptoms were reported by 232 students (16.6%) (Table 2). Severe and extremely severe anxiety symptoms were reported by 480 students (34.3%), while 420 (30.0%) students were exposed to moderate, severe, or extremely severe stress (Table 2).

The associations between following lifestyle habits of the students and mental health disorders were examined by a Pearson correlation, or by Spearman’s rho rank correlation—these correlations’ coefficients are shown in Table 3.

It was found that the symptoms of depression were more pronounced in students who more often consumed alcohol beverages and psychoactive substances during the pandemic (r = 0.16, *p* < 0.01, for both).

Anxiety symptoms were slightly more pronounced in students who slept longer during the night (r = 0.10, *p* < 0.01), consumed more alcoholic beverages, especially during the pandemic (r = 0.14, *p* < 0.01), as well as in students to whom social networks provided an adequate substitute for content that was denied (r = 0.15, *p* < 0.01).

Students who consumed significantly more alcoholic beverages (r = 0.16, *p* < 0.01), but also psychoactive substances (r = 0.19, *p* < 0.01), were under more stress, followed by those who had fewer friends in real life (r = −0.10, *p* < 0.01), as well as those who were not in an emotional relationship during the pandemic (r = 0.09, *p* < 0.01) (Table 3). Between male and female students, differences in correlations were shown in the occurrence of depressive (ro = 0.10, *p* < 0.01) and anxiety symptoms (ro = 0.12, *p* < 0.01), while the age of the respondents did not have any role.

After performing correlation tests, models for multiple linear regression analysis were created with those parameters that were shown to have a statistically significant degree of correlation with emphasized symptoms of depression, anxiety or stress (Table 4). Preliminary analysis proved that the assumptions of normality, linearity, (non)multicollinearity and homogeneity of variance were not violated.

Multiple linear regression analysis found that related to gender of respondents symptoms of depression and anxiety were significantly different (β = 0.08 and β = 0.11, respectively).

For more expressed depressive symptoms, the “most responsible” parameters were the consumption of alcoholic beverages (β = 0.10) and psychoactive substances (β = 0.11) compared to the period before the COVID-19 pandemic. Alcohol consumption (β = 0.11), as well as the use of social networks as an adequate substitute for deprived content during the pandemic, played a major role in anxiety symptoms (β = 0.13). The consumption of alcoholic beverages compared to the period before the COVID-19 pandemic (β = 0.19) and the general use of psychoactive substances (β = 0.21) were most “responsible” for the increased stress levels.

More frequent consumption of alcoholic beverages during the pandemic, as well as general consumption of psychoactive substances, proved to be a significant predictor (*p* < 0.01) in all three models: β = 0.19 and β = 0.14 for depression symptoms; β = 0.11 and β = 0.06 for anxiety; and β = 0.19 and β = 0.21 for increased stress, respectively (Table 4).

Multiple linear regression did not single out the emotional discourse as a significant factor for the development of anxiety symptoms during pandemic (Table 4).

Finally, it should be highlighted that these three observed models explain 7.2% of the variance in the occurrence of depression symptoms, 7.6% of the variance of anxiety symptoms, and 8.3% of the variance of emphasized stress (Table 4).

## 4. Discussion

In our research, depressive symptoms were more pronounced in students who more often consumed alcohol and psychoactive substances during the pandemic.

Research from Saudi Arabia suggests that respondents’ level of depression during the pandemic was most significantly related to the severity of their insomnia, their baseline intolerance of uncertainty, and whether they used problem or emotion coping strategies; the same applies to symptoms of anxiety and stress [24].

However, anxiety symptoms in our subjects were somewhat more pronounced in those who slept longer at night, consumed more alcoholic beverages, especially during the pandemic, as among students to whom social networks provided an adequate substitute for content denied during the pandemic. Researchers from the US have found that one of the main mechanisms linking the COVID-19 pandemic and mental illness may be cognitive concerns and behavioral changes, particularly due to reduced outdoor activities and limited social interactions, such as concerns about going out in public, avoiding eating in restaurants, going to stores less and more online shopping [25]. A drop in general mood as well as health-related behaviors caused by COVID was also observed. Follow-up analyses of the wellness index showed declines in each of the following five behaviors after COVID-19: minutes of exercise, diet quality, sleep, and hydration along with increases in screen time [26]. In younger age groups, the length of exercise (weekly) was shown to be a protective factor against depression during the pandemic, while poor sleep had a negative impact on the appearance of symptoms of depression and anxiety [27].

In Brazil, a meta-analysis proved that medical students have a rather permanent tendency to experience anxiety, stress and worries [28]. In Saudi Arabia, medical students also report more stress than non-medical students [29].

Under greater stress in this research were students who during the pandemic consumed significantly more alcoholic beverages and psychoactive substances than those who had fewer friends in real life, as well as those who were not in an emotional relationship during the pandemic. In Saudi Arabia, the sociodemographic factor associated with higher stress scores is marriage, divorce, and widowhood [24]. History of contacts with a suspected or confirmed case of COVID-19 was also a significant factor for the manifestation of all three types of symptoms in respondents in Saudi Arabia but also in China where infecting a relative or friend was moderately to highly significantly associated with an increased risk of three mental health problems [24,29,30].

Between male and female students in our country, differences in correlation were shown in the increased levels of depressive and anxiety symptoms. Age played no role in either case. For all three subscales of the DASS-21, female gender was associated with a higher level of negative emotions in respondents from Saudi Arabia, while increasing age was associated with a lower level of negative emotions [24,31]. Being married, as well as being a parent, were protective factors against psychological distress [31]. Americans under the age of 50 also had lower scores for all three scales of the DASS 42 during the pandemic [32]. In addition, in China, for respondents under the age of 50, who have lower levels of income and a personal history of psychiatric disorders, a stronger association with symptoms of depression, anxiety, insomnia and acute stress was found. In this case, male participants and the unmarried showed a significantly higher risk of depression, insomnia and symptoms of acute stress [32,33,34,35,36]. It is similar in Pakistan [34].

In a review that included 34 studies from more than 10 different countries, the most commonly reported mental health symptoms after a COVID-19 episode were anxiety, with prevalence ranging from 6.5% to 63% [37]. The second most common psychological symptom was depression, with a prevalence between 4% and 31% at follow-up times longer than 1 month after COVID-19. One third of patients in Italy and 41.3% of patients in Iran had both depression and anxiety after hospital discharge. Another common mental health problem after COVID-19 was PTSD (Post-traumatic stress disorder), with prevalence ranging from 12.1% to 46.9% [37].

An analysis of different population groups in India and Bangladesh revealed that students had higher rates of depression, anxiety and stress than other people, as well as health workers. Associated risk factors for mental health problems were gender, age, neighborhood, family size, monthly income, educational status, marital status, physical activity, smoking, alcohol use, fear of COVID-19, presence of chronic disease, unemployment status, and news exposure and social media related to COVID-19 [38].

Students were identified as a particularly high-risk group in a similar study conducted in a Saudi Arabian population. In addition, female gender, younger age group, unmarried/divorced marital status, lower education, lower income, unemployment status, living in a small family and living with the elderly were also recognized as risk factors for mental health problems during the COVID-19 pandemic. When it comes to behavioral and health factors of predictive importance, smoking, lack of physical activity, reduced immune status and lower resistance, chronic health problems and a history of psychiatric diseases but also factors related to COVID-19 itself, namely, lack of knowledge, fear and concern, infection or death of a family member or friend, restriction of movement, quarantine, and confirmed or suspected COVID-19 infection [39].

A meta-analysis of 66 papers, including 221,970 subjects, showed that non-infectious patients with chronic diseases, quarantined persons and patients with COVID-19 had a higher risk of depression and anxiety compared to other population groups [40].

Nevertheless, a large national representative study in Serbia reported that there were no significant differences between pre-pandemic and during COVID-19 pandemic symptoms of depression and anxiety, so called COVID-related stressors, among the respondents [41]. Similar to that, another study in Norway reported stable levels of mental disorders but also suicidal ideation and suicide deaths during the pandemic compared to pre-pandemic levels [42].

However, a recent study from Croatia has showed that the prevalence of depression (50.8%), anxiety (50.9%), and stress (49.9%) symptoms were much higher during the pandemic. In addition, female respondents had significantly higher levels of these symptoms [43]. In addition, the findings of Winkler et al. suggest that public mental health has not returned to pre-COVID-19 levels [44]. There was also a significant increase in weekly binge drinking behaviors, particularly highlighted during lockdowns. Younger adults, especially students, displayed a disproportionally high prevalence of mental disorders, which were highest during the second wave of the pandemic [44].

A systematic review and meta-analysis of over 2 million people performed by Delpino et al., including 194 studies, found the general prevalence of anxiety was 35.1% and that the prevalence in low and middle-income countries was similar compared to high-income countries. According to them, one in three adults was living with anxiety disorder during the worldwide COVID-19 pandemic [45].

According to Ochnik et al. who conducted a study on depression and anxiety among university students in nine countries, the highest depression and anxiety risk occurred in Turkey, the lowest depression rates were in Czech Republic and the lowest anxiety was marked in Germany [46].

### Strengths and Limitations

In accordance with these findings, it should also be mentioned that the strengths of this study are certainly a good sample, then the diversity of the faculties and reliable instruments, while the limitations would refer to having only one university in which students were investigated and the self-assessment as an approach itself.

## 5. Conclusions

During the COVID-19 pandemic, symptoms of depression, anxiety and stress were significantly increased among university students. More frequent consumption of alcoholic beverages and psychoactive substances proved to be a significant predictor of these increased symptoms, which indicates that this is how those students coped with the problems they were going through during this period. The results of our research lead us to the conclusion that in emergency situations, no matter how drastic they may be, we should turn to healthy lifestyles and social support from the immediate environment as the most important health determinants and protective factors that we can influence. The combined forces of health professionals and different medical sectors are necessary to develop strategies and tools that will respond to the health needs of students and other young people in order to protect their mental health and overall well-being.

## Figures and Tables

**Table 1 medicina-59-00180-t001:** Lifestyle habits of the surveyed students corresponding to COVID-19 pandemic.

	Total		Female		Male	
	Mean	SD	Mean	SD	Mean	SD
How many hours by day you spend on social networks?	3.23	1.97	3.44	2.30	3.02	1.56
How much do you sleep at night?	7.76	3.38	7.95	4.66	7.57	1.19
How often do you consume alcohol beverages?	2.56	1.04	2.37	1.03	2.73	1.01
Compared to the period before the COVID-19 pandemic is it:	1.87	0.54	1.89	0.53	1.86	0.54
Less (*n*, %)	301	21.5%	139	20.1%	162	22.9%
Same (*n*, %)	975	69.6%	491	71.0%	484	68.4%
More (*n*, %)	124	8.9%	62	9.0%	62	8.8%
How often do you consume psychoactive substances?	1.21	0.71	1.09	0.47	1.33	0.86
Compared to the period before the COVID-19 pandemic is it:	1.95	0.27	1.98	0.26	1.92	0.27
Less (*n*, %)	89	6.4%	32	4.6%	57	8.1%
Same (*n*, %)	1295	92.5%	644	93.1%	651	91.9%
More (*n*, %)	16	1.1%	16	2.3%	0	0.0%
How many close friends would you say that you have in real life?	4.72	3.77	3.99	2.38	5.43	4.64
Are you in a relationship? (*n*, %)	888 NO	63.4%	384 NO	55.5%	504 NO	71.2%
	512 YES	36.6%	308 YES	44.5%	204 YES	28.8%
How long have you been in a relationship, or single?	1.30	2.16	1.61	2.23	1.00	2.04
Were you under home isolation during the COVID-19 pandemic due to infection or contact with an infected person? (*n*, %)	756 NO	54.0%	408 NO	59.0%	348 NO	49.2%
	644 YES	46.0%	284 YES	41.0%	360 YES	50.8%
To what extent have social networks provided you with an adequate replacement for content that was denied during the pandemic?	5.00	2.84	5.17	2.76	4.84	2.91

**Table 2 medicina-59-00180-t002:** The scores of Depression, Anxiety, and Stress (DASS 42).

Symptom Levels	Depression			Anxiety			Stress		
	Total	Female	Male	Total	Female	Male	Total	Female	Male
Normal (*n*, %)	608	352	256	584	344	240	684	364	320
	43.4%	25.1%	18.3%	41.7%	24.6%	17.1%	48.9%	26.0%	22.9%
Mild (*n*, %)	228	84	144	108	60	48	184	72	112
	16.3%	6.0%	10.3%	7.7%	4.3%	3.4%	13.1%	5.1%	8.0%
Moderate (*n*, %)	260	116	144	228	68	160	272	144	128
	18.6%	8.3%	10.3%	16.3%	4.9%	11.4%	19.4%	10.3%	9.1%
Severe (*n*, %)	72	40	32	236	92	144	144	48	96
	5.1%	2.9%	2.3%	16.9%	6.6%	10.3%	10.3%	3.4%	6.9%
Extremely severe (*n*, %)	232	100	132	244	128	116	116	64	52
	16.6%	7.1%	9.4%	17.4%	9.1%	8.3%	8.3%	4.6%	3.7%

**Table 3 medicina-59-00180-t003:** Correlations between examined lifestyle habits of students with the symptoms of depression, anxiety, and stress.

	Depression	Anxiety	Stress
Gender	0.10 **	0.12 **	0.04
*p*	0.00	0.00	0.09
Age	0.04	0.04	0.02
*p*	0.10	0.13	0.34
How many hours by day do you spend on social networks?	−0.01	0.04	0.01
*p*	0.74	0.12	0.80
How much do you sleep at night?	−0.01 *	0.10 **	0.03
*p*	0.04	0.00	0.29
How often do you consume alcohol beverages?	−0.01	0.14 **	0.06 *
*p*	0.96	0.00	0.04
Compared to the period before the COVID-19 pandemic	0.16 **	0.13 **	0.16 **
*p*	0.00	0.00	0.00
How often do you consume psychoactive substances?	0.16 **	0.09 **	0.19 **
*p*	0.00	0.01	0.00
Compared to the period before the COVID-19 pandemic	0.01	0.04	−0.01 *
*p*	0.66	0.10	0.01
How many close friends would you say that you have in real life?	−0.06 *	−0.06 *	−0.10 **
*p*	0.02	0.01	0.00
Are you in a relationship?	−0.03	0.02	−0.04
*p*	0.23	0.53	0.12
How long have you been in a relationship?	−0.09 **	−0.02	−0.08 **
*p*	0.01	0.56	0.01
How long have you been single?	0.02	0.05 *	0.09 **
*p*	0.58	0.04	0.00
Were you under home isolation during the COVID-19 pandemic due to infection or contact with an infected person?	0.04	0.03	0.056 *
*p*	0.13	0.25	0.03
To what extent have social networks provided you with an adequate replacement for content that was denied during the pandemic?	−0.03	0.15 **	0.04
*p*	0.23	0.00	0.19

Notes: ** Correlation is significant at the 0.01 level, and * at the 0.05 level.

**Table 4 medicina-59-00180-t004:** Multiple linear regression analysis of depression, anxiety, and stress scores as continuous dependent variables with observed parameters.

Examined Parameters in Each of Three Models Observed	Unstandardized Coefficients		Standardized Coefficients	T	*p*	95% CI	
	B	Std. Error	β			Lower	Upper
DEPRESSION				R^2^ = 0.072; F = 19.093, df = 6, *p* < 0.001			
Constant	1.359	0.182		7.476	0.000	1.003	1.716
Gender	0.221	0.080	0.075	2.766	0.006	0.064	0.378
LF2	−0.031	0.011	−0.071	−2.722	0.007	−0.054	−0.009
LF3.1	0.525	0.072	0.190	7.255	0.000	0.383	0.667
LF4	0.295	0.055	0.141	5.341	0.000	0.187	0.403
LF5	−0.029	0.010	−0.074	−2.799	0.005	−0.049	−0.009
LF6.1	−0.063	0.018	−0.092	−3.511	0.000	−0.098	−0.028
ANXIETY				R^2^ = 0.076; F = 15.433, df = 8, *p* < 0.001			
Constant	0.764	0.213		3.588	0.000	0.346	1.181
Gender	0.350	0.087	0.112	4.012	0.000	0.179	0.521
LF2	0.037	0.012	0.079	3.037	0.002	0.013	0.061
LF3	0.144	0.041	0.095	3.541	0.000	0.064	0.224
LF3.1	0.322	0.077	0.110	4.207	0.000	0.172	0.473
LF4	0.139	0.059	0.063	2.359	0.018	0.023	0.255
LF6	−0.035	0.011	−0.083	−3.152	0.002	−0.056	−0.013
LF6.2	0.011	0.007	0.044	1.642	0.101	−0.002	0.025
LF8	0.073	0.014	0.133	5.093	0.000	0.045	0.101
STRESS				R^2^ = 0.083; F = 26.229, df = 5, *p* < 0.001			
Constant	1.949	0.287		6.782	0.000	1.385	2.512
LF3	0.016	0.034	0.012	0.468	0.640	−0.051	0.083
LF3.1	0.473	0.065	0.188	7.250	0.000	0.345	0.601
LF4	0.399	0.050	0.209	7.945	0.000	0.300	0.497
LF4.1	−0.547	0.133	−0.109	−4.108	0.000	−0.808	−0.286
LF5	−0.028	0.009	−0.079	−3.051	0.002	−0.047	−0.010

Notes: β—Beta coefficient in regression; CI—confidence interval; R^2^—coefficient of determination. LF1—How many hours by day do you spend on social networks? LF2—How much do you sleep at night? LF3—How often do you consume alcohol beverages? LF3.1—How often compared to the period before the COVID-19 pandemic? LF4—How often do you consume psychoactive substances? LF4.1—How often compared to the period before the COVID-19 pandemic? LF5—How many close friends would you say that you have in real life? LF6—Are you in a relationship? LF6.1—How long have you been in a relationship? LF6.2—How long have you been single? LF7—Were you under home isolation during the COVID-19 pandemic due to infection or contact with an infected person? LF8—To what extent have social networks provided you with an adequate replacement for content that was denied during the pandemic?

## Data Availability

Data is unavailable due to privacy or ethical restrictions.

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
