# Peer review of "Coping with Symptoms of Mental Health Disorders among University Students during the COVID-19 Pandemic in Relation to Their Lifestyle Habits"

_medicina, 2023, doi:10.3390/medicina59010180_

Round 1
Reviewer 1 Report
Dear authors
Congratulations on this significant work.
Some comments would be:
- Add the Cronbach alpha score and internal consistency, validity and reliability of the scales used.
- Add sample size calculation
- Add IRB number and more details regarding controlling biases in sample selection, and throughout the procedure.
- Report confidence intervals next to reported means for better understanding of the resulting findings
- It seems that the regression models showed that the gender and lifestyle habits predicted the outcome variables in low percentages as shown by the r squared. This should be highlighted in the results and recommendations section.
Author Response
Dear colleague,
Thank you very much for the selfless attention you have given to our work.
We have given our best to adequately respond to your comments and suggestions.
If there are any other suggestions or remarks, please let us know to make our manuscript even more better.
In the revised text, we marked in blue all the changes that we have made.
- Add the Cronbach alpha score and internal consistency, validity and reliability of the scales used.
Internal consistency of the entire DASS 42 scale expressed by Cronbach's alpha coefficient was α = 0.88.
- Add sample size calculation
During this period, there were a total of 24,047 students at the University of Niš, and according to the valid statistical formula, for a confidence level of 95%, the minimum sample size calculated was 379 students. This confirms that our randomized sample was very representative.
- Add IRB number and more details regarding controlling biases in sample selection, and throughout the procedure.
The Ethics Committee of the Faculty of Medicine of the University of Niš approved the study (No. 12-6647-2/6).
- Report confidence intervals next to reported means for better understanding of the resulting findings
Actually, in Table 1. we have only calculated descriptive measures for each of the examined parameters.
- It seems that the regression models showed that the gender and lifestyle habits predicted the outcome variables in low percentages as shown by the r squared. This should be highlighted in the results and recommendations section.
It should be highlighted that these three observed models explain 7.2% of the variance in the occurrence of depression symptoms, 7.6% of the variance of anxiety symptoms, and 8.3% of the variance of emphasized stress (Table 4).
Reviewer 2 Report
Here you can find my appraisal and commentaries .
I found the introduction very brief and concise. Reading the introduction, l was expecting to find an exhaustive literature review of the studies about Covid -19 pandemic and related psychiatric disorders. Indeed, authors can add more specific information about students and Covid -19 pandemic. Most of the studies performed during the pandemic and lockdowns, used online questionnaires. Most of them have been carried out by universities and it is possible that the academic population ( including students) has been used.
Please add more information about.
At the end of the introduction,the aims or the hypotheses need to be more explicit and supported by a previous review of the literature.
According to me, the authors could reformulate the introduction.
In the methods, the authors collected data between December 2021 and February 2022. I suppose that during that period, most of the European countries started their own vaccination programs and some of the restrictions have been revoked. Please add more information about this specific choice. Moreover, l invite the authors to explain this in the text. Then , the authors could also add info about the restrictions in Serbia during COVID .
Authors used the DASS test ( not a device, but a tool..it is a psychometric test, not a plethysmograph). The authors did not specify if they used a validated version of the DASS for the Serbian population. I did not find this information in the text or in the reference list. However, it is important to use tests that were previously adapted and validated in another language different from the original one.
In the data analysis, the authors need to be more clear about the use of parametric or non-parametric tests.
Likert scale: correct please. Likert scale (7 - points) the 5 is equal to zero. Likert like scale has a min and a max value.
The first column of table 4 is not intelligible.
In the discussion, the limitations should be added in a specific paragraph. Moreover, l advice to add references about behavioral alterations ( addiction) in social and economic crises or wars.
Author Response
Dear reviewer,
Thank you for the detailed analysis of our work and the attached corrections and suggestions.
We tried our best to improve the manuscript according to your appraisal and commentaries. Also, we hope that we have removed all possible misunderstandings.
In the revised text of the paper, the changes and additions we have made accordingly are marked in red.
If you have any other objections, please let us know and we will certainly act accordingly.
I found the introduction very brief and concise. Reading the introduction, l was expecting to find an exhaustive literature review of the studies about Covid -19 pandemic and related psychiatric disorders. Indeed, authors can add more specific information about students and Covid -19 pandemic. Most of the studies performed during the pandemic and lockdowns, used online questionnaires. Most of them have been carried out by universities and it is possible that the academic population ( including students) has been used.
Please add more information about.
At the end of the introduction,the aims or the hypotheses need to be more explicit and supported by a previous review of the literature.
According to me, the authors could reformulate the introduction.
First of all, as you can see, the whole Introduction section is now upgraded, according to your advices. Hopefully, you will approve that some of the remarks which you have also mentioned later (for other sections of the manuscript) are included now in this part.
In the methods, the authors collected data between December 2021 and February 2022. I suppose that during that period, most of the European countries started their own vaccination programs and some of the restrictions have been revoked. Please add more information about this specific choice. Moreover, l invite the authors to explain this in the text. Then , the authors could also add info about the restrictions in Serbia during COVID.
On the one hand, it should be noted that vaccination coverage among the student population in Serbia was at a low level. Nevertheless, despite that vaccine campaign, most of the restrictions were still in place in the study period or just before the study period. In the end, in our work we have dealt primarily with topics that had no direct connection with this (which would have drawn us into a completely different research.
Authors used the DASS test (not a device, but a tool..it is a psychometric test, not a plethysmograph). The authors did not specify if they used a validated version of the DASS for the Serbian population. I did not find this information in the text or in the reference list. However, it is important to use tests that were previously adapted and validated in another language different from the original one.
The full DASS scale was translated into Serbian by Dr. Zoran Protulipac, clinical psychologist and translator, in consultation with the original authors of the scale. The translation is available on the official website of the instrument:
http://www2.psy.unsw.edu.au/Groups/Dass/Serbian/DASS-SER.pdf.
Some further informations about this are available at:
Popov, S., & Sokić, J. (2022). Psychometric Characteristics of a Serbian Translation of Unconditional Self-Acceptance Questionnaire and Development of a Short Form. Psihologija, 55(1), 107–122. doi: https://doi.org/10.2298/PSI200820005P
and
Jovanović V, Gavrilov-Jerković V, Žuljević D, Brdarić D. Psychometric evaluation of the Depression Anxiety Stress Scales in a Serbian student sample. PSIHOLOGIJA, 2014, Vol. 47(1), 93–112. DOI: 10.2298/PSI1401093J
In addition, the psychometric properties of the DASS–42 and DASS–21 have been extensively tested in both clinical (e.g., Gloster et al., 2008; Page, Hooke, & Morrison, 2007) and nonclinical samples (e.g., Crawford & Henry, 2003; Sinclair et al., 2012). Almost all research shows that DASS scales have adequate reliability, with internal consistency that most often ranges from 0.80 to 0.95.
In the data analysis, the authors need to be more clear about the use of parametric or non-parametric tests.
Actually we have only used descriptive tests, multiple linear regression analysis, and correlation tests in our statistical analysis.
Likert scale: correct please. Likert scale (7 - points) the 5 is equal to zero. Likert like scale has a min and a max value.
We have actually used 6-point Likert scale, which was, by our opinion, most suitable for measuring the frequency of alcohol and psychoactive substances consuming.
The first column of Table 4. is corrected, and now it is:
Examined parameters in each of three models observed
Reviewer 3 Report
It is a well presented manuscript, plagiarism is not detected in any of its forms or sections. I believe that it is a topic that will be of interest to readers and that it leaves open an interesting topic for discussion about the effects of the isolation associated with the pandemic.
I would only request to know if the data were analyzed with respect to their distribution, if normal, please specify it and by which test normality was determined, otherwise, explain it and analyze if the use of the statistical tests used are adequate.
Excellent work.

Author Response
Dear reviewer and dear colleague,
Thank you very much for the excellent marks you gave to our work.
The addition we have made in accordance with your observations is marked in purple in the text of the revised paper.
It is a well presented manuscript, plagiarism is not detected in any of its forms or sections. I believe that it is a topic that will be of interest to readers and that it leaves open an interesting topic for discussion about the effects of the isolation associated with the pandemic.
I would only request to know if the data were analyzed with respect to their distribution, if normal, please specify it and by which test normality was determined, otherwise, explain it and analyze if the use of the statistical tests used are adequate.
Excellent work.
Preliminary analysis prove that the assumptions of normality, linearity, (non)multicollinearity and homogeneity of variance were not violated.